# PNPLA6/NTE, an Evolutionary Conserved Phospholipase Linked to a Group of Complex Human Diseases

**DOI:** 10.3390/metabo12040284

**Published:** 2022-03-24

**Authors:** Doris Kretzschmar

**Affiliations:** Oregon Institute of Occupational Health Sciences, Oregon Health and Science University, Portland, OR 97239, USA; kretzsch@ohsu.edu

**Keywords:** Neuropathy Target Esterase, Swiss-Cheese, organophosphate-induced delayed neuropathy, hereditary spastic paraplegia, phosphatidylcholine, Protein kinase A

## Abstract

Patatin-like phospholipase domain-containing protein 6 (PNPLA6), originally called Neuropathy Target Esterase (NTE), belongs to a family of hydrolases with at least eight members in mammals. PNPLA6/NTE was first identified as a key factor in Organophosphate-induced delayed neuropathy, a degenerative syndrome that occurs after exposure to organophosphates found in pesticides and nerve agents. More recently, mutations in PNPLA6/NTE have been linked with a number of inherited diseases with diverse clinical symptoms that include spastic paraplegia, ataxia, and chorioretinal dystrophy. A conditional knockout of PNPLA6/NTE in the mouse brain results in age-related neurodegeneration, whereas a complete knockout causes lethality during embryogenesis due to defects in the development of the placenta. PNPLA6/NTE is an evolutionarily conserved protein that in *Drosophila* is called Swiss-Cheese (SWS). Loss of SWS in the fly also leads to locomotory defects and neuronal degeneration that progressively worsen with age. This review will describe the identification of PNPLA6/NTE, its expression pattern, and normal role in lipid homeostasis, as well as the consequences of altered NPLA6/NTE function in both model systems and patients.

## 1. Introduction

Changes in PNPLA6/NTE have now been shown to be involved in neurodegenerative symptoms in species ranging from flies to humans. That mutations in this protein can cause neurodegeneration and locomotion deficits was first shown in *Drosophila* by the identification of point mutations in the *swiss cheese* (*sws*) gene, the fly homolog of PNPLA6/NTE, that lead to progressive neuronal degeneration during adulthood [1]. Human PNPLA6/NTE, whose activity has long been associated with a neuropathy caused by organophosphate poisoning, was subsequently cloned and found to be homologous to fly SWS [2]. In recent years, mutations in PNPLA6/NTE have also been linked with a number of rare inherited diseases with clinical symptoms that can include spastic paraplegia, ataxia, hypogonadism, and chorioretinal dystrophy [3,4]. PNPLA6/NTE proteins contain several conserved domains: (1) a phospholipase domain that provides its catalytic activity as esterase and contains the binding sites for the organophosphates, (2) a transmembrane spanning domain that localizes the majority of the protein to the membranes of the endoplasmic reticulum, and (3) domains that have homology with the regulatory subunits of Protein Kinase A (PKA). Specifically, this domain includes three nucleotide binding sites and a region with homology to canonical regulatory subunits of PKA required for binding to the catalytic subunits of PKA [2,5]. Whereas the phospholipase function as been confirmed in several species [6,7,8], the PKA regulatory function has so far only been confirmed in *Drosophila,* although mammalian PNPLA6/NTE can bind to the fly PKA catalytic subunit [9]. In this review, we further describe the functions of PNPLA6/NTE, its expression pattern, and the symptoms that are caused when mutated or inhibited in animal models or humans. 

## 2. PNPLA6/NTE and Organophosphate-Induced Delayed Neuropathy

Organophosphate-induced delayed neuropathy or OPIDN was first described in 1930 after a poisoning epidemic in the southern United States, when thousands of people were paralyzed after consuming a beverage called Jamaica Ginger that contained the organophosphorus compound tri-ortho-cresyl phosphate (TOCP) [10]. In search for a molecular target, an esterase activity was identified in chicken brains that was eventually called Neuropathy Target Esterase or NTE [11,12,13]. The NTE gene was cloned in 1998 [2] and was later found to correspond to PNPLA6; accordingly, we will refer to this protein as PNPLA6/NTE. OPIDN occurs when an organophosphorus compound (OP) binds to a serine residue within the catalytic domain of PNPLA6/NTE and inhibits its activity against an artificial substrate (often used is phenyl valerate) by at least 70% [14]. However, inhibition of its esterase activity alone is not sufficient, because some OPs can inhibit PNPLA6/NTE very efficiently but not cause OPIDN. Compounds of this type are therefore called non-neuropathic and they can even protect against effects of neuropathic OPs [15]. Neuropathic OPs have been shown to induce the so-called aging reaction, in which a side group of the OP is lost, resulting in a net negative charge and the subsequent covalent binding of the OP to PNPLA6/NTE [16] (Figure 1). It has therefore been suggested that OPIDN is due to two effects: a loss of PNPLA6/NTE function due to the inhibition of its phospholipase function and a dominant gain-of function effect due to the aging reaction. Although large outbreaks of OPIDN have not been described more recently, single cases or small outbreaks occur regularly [17]. Neuropathic OPs can still be found in many commercial products, including pesticides, insecticides, flame retardants, and lubricants and they are also active compounds in warfare agents such as sarin [18,19]. A detailed review of neurotoxic compounds, their chemical reactions and interactions with PNPLA6/NTE has recently been published by Richardson and colleagues [19].

As the name implies, the onset of symptoms is delayed in OPIDN, typically developing 2–4 weeks after exposure [20,21]. Clinical signs include tingling and pricking of the hands and feet followed by sensory loss and a progressive muscle weakness that culminates in ataxia and paralysis [22]. Histologically, OPIDN is characterized by a Wallarian-type axonopathy that primarily affects the lower extremities but can include upper extremities in more severe cases. This is accompanied by demyelination that appears to result from Schwann cells failing to synthesizing new myelin [23,24]. In mild cases, patients can recover, although symptoms may linger for some time [18,20,25,26]. More severe cases lead to immobility and, in one case it has even resulted in the death of the affected individual [3,27].

## 3. PNPLA6/NTE Is an Evolutionarily Conserved Protein with Several Functional Domains

PNPLA6/NTE is an evolutionarily conserved protein with orthologues found in organisms ranging from *Drosophila* to humans, and proteins with homology to the C-terminal region of PNPLA6/NTE can even be found in bacteria [2,5]. In humans, four canonical isoforms generated by alternative splicing have been described, the original isoform identified is now designated isoform 2. This isoform contains 1327 amino acids, slightly shorter than the longest isoform (isoform 4) with 1375 amino acids. Alternative splicing is also predicted to produce four isoforms in mice, whereas only one canonical PNPLA6/NTE is found in Zebrafish (UniProt). The *Drosophila* gene encodes for two alternative isoforms, both with 1425 amino acids. In addition, several smaller potential isoforms have been predicted in all of these species.

By analyzing the sequences of PNPLA6/NTE proteins from different species, four conserved domains can be identified (Figure 2). The highest conservation between species is found in a C-terminal region of PNPLA6/NTE and this is also the region that shows homology to patatin, a lipase found in plants. It was therefore assumed that this region contains the esterase catalytic domain, which was confirmed by Atkins and Glynn [28]. In their studies, they showed that a 489 amino acid long recombinant fragment (named NEST) containing this region has esterase activity against the substrate phenyl valerate. However, the activity of NEST alone is lower than when using the full-length protein [29]. They also showed that a mutation in a serine residue in this region (Ser1014) abolished catalytic activity. This serine also provides the binding site for OPs [30]. Furthermore, the catalytic activity requires two aspartic acids in this region whereby Asp1134 and Ser1014 form the active catalytic center [28,30]. The importance of the active site serine was also confirmed in vivo in the *Drosophila* model. As discussed in more detail below, flies with a mutation in the *swiss-cheese* (*sws*) gene, encoding the fly orthologue of PNPLA6/NTE show neuronal degeneration and a loss of esterase activity against phenyl valerate. Whereas expressing a normal wildtype SWS in these flies can restore esterase activity and prevent neurodegeneration, a construct with a mutation in the active site serine (Ser985Asp) can do neither [9,31].

The N-terminal half of SWS contains a region with homology to the regulatory subunit of Protein kinase A (PKA). PKA is a complex of two catalytic subunits and two regulatory subunits and the binding of the regulatory subunits to the catalytic subunits is mediated by this domain (PKA-BD in Figure 2). When cAMP binds to the canonical regulatory subunits, the catalytic subunit is released and becomes active [32,33]. That this is a functional domain was again shown in *Drosophila* by expressing a SWS construct with a mutation in this domain (SWS^Arg133Ala^). In comparison to the wildtype construct, SWS^Arg133Ala^ only partially rescued neuronal degeneration and surprisingly it could also not restore esterase activity in *sws* mutant flies [9]. Whether this is due to a direct effect of the mutation on catalytic activity or is caused by changes in the structure of the protein that interfere with activity still needs to be determined. To assess whether the PKA-BD domain is needed to interact with a PKA catalytic subunit, the wildtype and mutant constructs were used in a two-hybrid experiment [9]. This experiment revealed that SWS specifically binds to PKA-C3, one of the three catalytic subunits identified in *Drosophila,* and that binding is mediated by the PKA-BD domain. Furthermore, *sws* mutant flies showed an increase in PKA activity, suggesting that as with canonical subunits, the binding of SWS to PKA-C3 inhibits its activity. Overexpressing PKA-C3 in *sws* mutant flies aggravated the degenerative phenotype, suggesting that disrupting the normal interactions between SWS and PKA-C3 has functional consequences for the nervous system. Although an interaction of PNPLA6/NTE and the mammalian orthologues of PKA-C3 (Pkare/PRKX in mouse and humans) has so far not been investigated, the finding that mouse PNPLA6/NTE can bind to *Drosophila* PKA-C3 in two-hybrid studies [34], suggests that the function of this domain might be conserved in mammals. 

As mentioned above, canonical regulatory subunits of PKA bind cAMP which then results in the release of the catalytic subunits. Interestingly, three cyclic nucleotide binding sites are also present in SWS and PNPLA6/NTE (cNMP, Figure 2), suggesting that PNPLA6/NTE function is regulated by cyclic nucleotide binding. Using HeLa cells (which do express PNPLA6/NTE), Chen and colleagues found that treatment with chemicals that increase cAMP levels resulted in an increase in PNPLA6/NTE levels and esterase activity [35]. In contrast, inhibiting adenylate cyclase and therefore reducing cAMP levels decreased PNPLA6/NTE levels and activity. These results suggest that cAMP regulates the phospholipase activity of PNPLA6/NTE in addition to regulating its binding to PKA catalytic subunits.

Lastly, structural analyses and modeling predict that PNPLA6/NTE proteins contain a single transmembrane domain (TM) at the N-terminus [30] and in *Drosophila*, SWS is exclusively detected in membrane fractions [9]. That this membrane localization is important for the esterase function was suggested by experiments showing that the NEST fragment described above loses catalytic activity in detergent containing extracts but regains it when incorporated into liposomes [28]. Furthermore, an PNPLA6/NTE construct lacking the TM has reduced esterase activity when expressed in COS cells [29]. In addition, the membrane localization of PNPLA6/NTE may be important for PKA activity because it has been shown that SWS anchors the PKA-C3 catalytic subunits to the membrane [9]. Together with the localization of PNPLA6/NTE to the endoplasmic reticulum (ER) described below, these results support a model whereby PNPLA6/NTE is inserted into ER membranes with the short N-terminal part residing in the ER lumen while the majority of the protein is on the cytoplasmic side. 

## 4. Phospholipase Function of PNPLA6/NTE

PNPLA6/NTE belongs to a family of hydrolases that react with different substrates such as phospholipids, triacylglycerols, and retinol esters [36]. PNPLA6/NTE acts as phospholipase, preferably hydrolyzing phosphatidylcholine (PC) and lysophosphatidylcholine (LPC) [2,37,38,39]. It has therefore been suggested that PNPLA6/NTE is able to produce glycerophosphocholine (GPC) from PC directly via cleavage of sn-1 and sn-2 bonds (similar to phospholipase B) or by generating GPC indirectly via LPC (Figure 3) [40]. 

The first indication that PNPLA6/NTE might act as a phospholipase came from the discovery that the activity of purified PNPLA6/NTE was increased in the presence of phospholipids [41,42]. This was further supported by experiments in cell culture, which revealed that overexpression of PNPLA6/NTE increased the levels of LPC, whereas knocking down PNPLA6/NTE reduced them [6,43]. Another study found that overexpression of the NEST fragment in cultured cells induced LPC hydrolysis activity, reduced PC levels, and changed the fluidity of membranes [44,45]. The phospholipase function was also confirmed in mice where the loss of PNPLA6/NTE caused an increase in PC [7]. Similarly, *sws* mutant flies showed an increase in PC and LPC whereas expression of additional SWS reduced PC (LPC was not measured) [7,8,31]. Inhibition of PNPLA6/NTE in primary neurons and astrocytes by neurotoxic organophosphates also reduced GPC levels [7,46] and hens treated with neuropathic TOCP showed an increase in both PC and LPC in addition to a reduction of GPC [47]. Other publications confirmed a decrease in GPC after treating hens or mice with TOCP, although they did not find an increase in PC or LPC [48,49]. These observations suggest that a reduction in PC and LPC levels is not a major factor in inducing OPIDN, whereas an increase in GPC might play a central role in this neurotoxic response. 

Interestingly, findings in *Drosophila* suggest that disruptions in the PKA-regulatory function may also play a key role in the development of OPIDN. Treating *Drosophila* with TOCP inhibited SWS activity and caused delayed locomotion deficits and neurodegeneration, similar to findings in other models [34]. Unexpectedly, overexpressing SWS in TOCP-treated flies did not prevent neurodegenerative responses but actually aggravated these phenotypes, a result that was confirmed by another group [50]. However, overexpressing PKA-C3 ameliorated the behavioral and degenerative phenotypes, while TOCP exposure was found to reduce PKA activity [34]. Together, these results suggest that a loss of PKA-C3 activity could play a critical role in OPIDN. 

## 5. Expression Pattern of PNPLA6/NTE

The first expression study of PNPLA6/NTE was performed on chicken brains by Glynn and colleagues which showed a widespread expression of the protein in the brain cortex, optic tectum, cerebellum, spinal cord, and dorsal root ganglia [51]. Immunostaining was especially strong in large neurons such as the Purkinje cells and neurons in the dorsal root ganglia. Although mostly restricted to cell bodies, PNPLA6/NTE was also detectable in the proximal region of axons in large neurons and it was found in the sciatic nerve after injury. Expression in the brain was confirmed by in situ hybridization which showed that the PNPLA6/NTE mRNA was also present outside the nervous system in the kidney, liver, and testis of chickens [52,53].

Analyzing the expression of the *Drosophila* SWS protein in the brain revealed a similar pattern as in chickens. In young adult flies (1d after eclosion), SWS is found in most or all neurons, primarily localized to the ER, but with aging, SWS expression became more restricted to large neurons [31]. In addition, it was found to be expressed in photoreceptors [8]. During larval development, expression in the central nervous system appeared to be absent, but SWS can be detected in peripheral nerves when driving CD8-GFP expression with a *sws* promoter construct [54]. SWS could also be found in a variety of glial cells, including ensheathing glia, which are similar to oligodendrocytes, and in the surface glia, which form the analogue to the blood–brain barrier and blood–nerve barrier in mammals [31,54]. Using the CD8-GFP reporter system, SWS also appeared to be expressed in the fat body (which fulfills similar functions as the liver, including lipid storage; [55,56]), intestine, and Malpighian tubes (an excretory and osmoregulatory system similar in function to the kidney [57]). In addition, SWS is expressed in the male reproductive system [58]. 

Similarly, in young mice PNPLA6/NTE is expressed in most or all neurons [5] whereas the pattern becomes more restricted with age to large neurons in the cortex, olfactory bulb, midbrain, thalamus, hypothalamus, pons, medulla oblongata, hippocampus, and the Purkinje cells in the cerebellum. During murine development, PNPLA6/NTE was first detected in the nervous system at stage 11 post coitum (pc) in the cranial and dorsal root ganglia. Although expression in these areas increased by stage 13 pc, PNPLA6/NTE remained low in the brain and spinal cord [5]. Furthermore, it was found in the developing and adult retina, including the horizontal, amacrine and photoreceptor cells, but not in retinal ganglion cells [8,59]. In the peripheral nervous system, PNPLA6/NTE was detectable in the sciatic nerve and it was upregulated after injury, as shown in chickens [60]. However, PNPLA6/NTE expression in the sciatic nerve was more prominent in glia, specifically in non-myelinating Schwann cells. While not detectable in immature and promyelinating Schwann cells, weak expression was seen around post-natal day 5 with increasing expression continuing throughout adulthood. Intracellularly, PNPLA6/NTE was enriched at the Schmidt–Lanterman incisures which form actin-rich cytoplasmic channels, and around the nucleus, consistent with a localization in the ER (as seen in neurons). Outside the nervous system, PNPLA6/NTE was first detected at stage 9 pc in the mesonephric and metanephric ducts, and by stage 13 pc also in the collecting duct system. At this stage, expression was also detected in the gut epithelium followed by expression in the olfactory, pharyngeal and tracheal epithelia at stage 15 pc [5]. In the adult, PNPLA6/NTE was found in many tissues, including the liver, spleen, prostate, and placenta [59].

Lastly, human PNPLA6/NTE has been detected in the developing cerebellum and hindbrain in Carnegie stage 19 embryos and in the retina, pituitary gland, and nasal epithelium at Carnegie stage 23 [59]. As in mice, human PNPLA6/NTE is also present in various tissues outside the nervous system, including kidney, liver, small intestine, and spleen. An in vitro analysis of COS cells transfected with GFP-tagged PNPLA6/NTE also suggested an enrichment of the human protein in the ER [29]. Together with the conserved structure of PNPLA6/NTE across species, these similarities in expression patterns suggest that PNPLA6/NTE may serve similar functions in species ranging from *Drosophila* to humans. 

## 6. Phenotypic Consequences Due to the Loss of PNPLA6/NTE in Model Systems

Due to the occurrence of OPIDN in humans, the first studies in animals focused on studying effects of OPs. Whereas rodents are relatively resistant to develop the clinical symptoms of OPIDN, adult hens have been established as a model early on [10] and it has been shown that they develop symptoms that are fairly similar to OPIDN in humans [61]. Treated hens have difficulties standing or walking and they reveal axonal degeneration and changes in myelin [62,63]. They have also been widely used to determine the effects of different OPs and to show that pre-treatment with non-neuropathic OPs can protect from the effects of neuropathic OPs [15,19,63]. 

Whereas rats do not show the severe phenotypes of chickens after OP treatment, they do reveal reduced locomotor activity and it takes them longer to cross a narrow beam [64]. In addition, they show axonal degeneration after administered a neuropathic OP which can also be prevented with pre-treatment with a non-neuropathic OP [65,66]. 

### 6.1. Drosophila Melanogaster

Performing a mutagenesis screen in *Drosophila*, Heisenberg and colleagues identified several mutants with structural brain defects, including five mutants showing severe neurodegeneration as seen by the formation of vacuoles in the brain [67]. A complementation analysis showed that they all affected the same gene which based on the appearance of the mutant phenotype was named *swiss-cheese* or *sws* (Figure 4). Cloning and sequencing revealed that two of the alleles contained point mutations in the *sws* gene [1]. Another mutation generated a stop codon early in the sequence, resulting in no protein being detectable in Western blots. This allele (*sws*^1^) therefore appears to result in a complete loss of SWS. The mutations in the remaining two alleles could not be identified. The phenotypic analyses showed that all *sws* alleles exhibit age-dependent neuronal degeneration due to apoptosis, glial cell death, and a reduced lifespan [1]. In addition, *sws* mutants develop locomotion deficits, disruptions in the ER and ER stress [31,68]. Specifically knocking down SWS in neurons also caused age-related neurodegeneration, locomotion deficits, reduced life span and a reduction of mitochondria in the CNS, whereas mitochondrial transport was reduced in the PNS [69]. Consistent with reduced mitochondrial function, these flies also showed an increase in reactive oxygen species, in addition to an accumulation of lipid droplets. A knockdown in all glial cells resulted in glial wrapping defects in the CNS and abnormal glial layers and fragmented glial nuclei in the PNS [54,70]. Furthermore, the pan-glial knockdown of SWS expression induced defects in neuronal transmission and locomotion, while more restricted knockdowns in particular glial subtypes revealed that SWS is specifically required in ensheathing and surface glia [70]. 

To determine the functional importance of the different domains identified in SWS, several constructs were tested for their ability to rescue the phenotypes seen in the *sws* mutant. Using a wildtype form of SWS, revealed that the neuronal degeneration could be rescued by selective expression in neurons, whereas the glial phenotype could be rescued by expression in glia, showing that SWS is autonomously required in both cell types [31]. In addition, expression in either neurons or glia restored esterase activity and for the neuronal rescue it was also demonstrated that it reduced the abnormal increase in PC seen in *sws* flies. In contrast, as mentioned above, a SWS construct with a mutation in the active site serine of the phospholipase domain did not restore esterase activity in *sws* mutants and it only partially rescued the neurodegeneration phenotype [9,31]. Similarly, a mutation in the PKA-BD (see Figure 2) domain only partially rescued the degenerative phenotype [9], suggesting that both domains contribute equally to the neuronal function of SWS. Although the PKA-BD mutant protein has an intact phospholipase domain, it could not restore the esterase activity in *sws* mutants; however, it doubled the level of esterase activity when expressed in the presence of wildtype SWS. These findings support the hypothesis that the PKA-regulatory function of SWS is required for its phospholipase activity, though not directly within the same molecule. In glia, the phospholipase function seems to play a more prominent role than the PKA-regulatory function, because the mutation in the active site serine completely failed to rescue the glial phenotypes in the glia-specific knockdown of SWS [70]. In contrast, expression of the PKA-BD mutant resulted in a partial rescue. Lastly, to confirm the functional conservation of SWS and mammalian PNPLA6/NTE, constructs encoding either the mouse or human PNPLA6/NTE were expressed in *sws* mutants, and both could rescue the behavioral and degenerative phenotypes, as well as restore esterase activity [31,71,72].

### 6.2. Danio Rerio

Effects of the loss of PNPLA6/NTE have also been studied in mammalian models. A knock-down in zebrafish embryos caused various abnormalities, with a curled tail being the most prominent phenotype [73]. In addition, the embryos showed defects in eye development, midbrain-hindbrain boundary abnormalities, and a reduced number of motor neurons with short and abnormally branched axons. Addressing the importance of the esterase domain, the authors co-injected PNPLA6/NTE-specific morpholinos with different PNPLA6/NTE mRNAs. Whereas expression of wildtype PNPLA6/NTE rescued the curly tail and motor neuron phenotype of the knock-down, three mRNAs that contained mutations in the esterase domain, including one with a mutation in the active site serine, did not. Similar experiments by Hufnagel et al. confirmed the curly tail phenotype in the PNPLA6/NTE knockdown and also demonstrated that wildtype PNPLA6 rescued this phenotype but PNPLA6/NTE with mutations in the esterase domain did not [59]. In combination, these studies provide further evidence that the esterase/phospholipase activity is important for the biological function of PNPLA6. 

### 6.3. Mus Musculus

Mice lacking PNPLA6/NTE show severe growth retardation and die around day 9 of embryonic development [74]. A histological analysis revealed defects in placental development (with no placental labyrinth forming), in addition to a breakdown of yolk sac circulation, leading to enlarged pericardia and dilated blood vessels in the embryo. A role early in embryonic development is also supported by findings that PNPLA6/NTE is expressed in mouse embryonic stem cells, with increased levels during differentiation, and that silencing caused changes in the expression of genes involved in neuronal development and the formation of the respiratory and vascular system [75,76]. Silencing of PNPLA6/NTE in a human pluripotent cell line induced similar changes in the transcriptome, suggesting that the function of PNPLA6/NTE during development is conserved in humans [77,78]. Interestingly, treatment with OPs did not induce these transcriptional changes, although it inhibited the esterase function, suggesting that this developmental function of PNPLA6/NTE is not dependent on its esterase activity [77,79]. Whereas the loss of PNPLA6/NTE and its developmental functions causes lethality during embryogenesis, heterozygous knock-out animals develop normally but have reduced esterase activity, increased susceptibility to certain OPs, and they appear hyperactive in open field tests [74,80]. A conditional, brain-specific knock-out did not affect development but these mice showed defects in motor coordination and neuronal degeneration when aged to 4.5 month [81]. Further analyses revealed a loss of Purkinje cells, disruptions of the ER, and abnormal reticular aggregates. In the spinal cord, axonal degeneration was first noticed in the distal parts of the longest spinal axons at 1 month of age, followed by axonal swelling that increased with aging in both, ascending and descending tracts [7]. These defects were accompanied by progressive hindlimb dysfunction, first detectable by clenching of the digits, and by 4 months the animals were unable to fully support their lower body when walking on a beam. Comparable axonal degeneration and progressive swelling of spinal cord axons was detected when treating animals with TOCP [7]. A glial-specific knock-out resulted in incomplete ensheathment of Remak fibers in the sciatic nerve whereas myelination was not affected [60], confirming that the loss of PNPLA6/NTE in glia also has functional consequences in mammals. Lastly, a selective knock-down of PNPLA6/NTE in testis demonstrated a role in the proliferation of spermatogonial stem cells and a reduction in sperm count was also observed when treating male mice with TOCP [82]. 

## 7. PNPLA6 in Human Disease

The first disease-causing mutations in PNPLA6/NTE were described in 2008 in two families with affected members showing progressive spastic paraplegia associated with distal upper and lower extremity wasting starting in childhood [83]. Electrophysiological studies and magnetic response imaging suggested that the locomotion defects were due to motor neuron neuropathy and spinal cord atrophy. Rainier and colleagues identified three mutations in these families, all localized within the esterase domain of PNPLA6/NTE. The mutations were recessive and whereas the patients in one family were homozygous, the affected individuals from the other family were compound heterozygotes. This inherited spastic paraplegia was consequently named NTE-related Motor Neuron Disorder or spastic paraplegia 39 (SPG39). Mutations in PNPLA6/NTE have now been identified in a few more spastic paraplegia patients and there are now also mutations outside the esterase domain [84] (Figure 5). In one family, the affected members also showed Parkinsonism in addition to spastic paraplegia [85,86], suggesting that the pathology can vary. In fact, it has now been shown that mutations in PNPLA6/NTE can cause a wide spectrum of neurological disorders with overlapping symptoms (Table 1), which have now been categorized as PNPLA6-related disorders [3,59,84,87]. 

One of them is Boucher–Neuhäuser syndrome, which is characterized by delayed puberty, spinocerebellar ataxia, and in some patients, chorioretinal dystrophy. Mutations in PNPLA6/NTE as the underlying cause for Boucher–Neuhäuser syndrome were first described in 2014 and there are now more than a dozen identified pathogenic mutations [84]. As in the case of SPG39, the mutations are recessive and affected individuals are either homozygous for the mutation or compound heterozygotes. Disease-causing mutations in these patients are mostly, but not exclusively, found in the esterase/phospholipase domain although a few are in the cyclic nucleotide binding sites [88,89,90,91,92,93,94,95,96,97,98]. The onset of symptoms varies widely and although a case of a one-year-old child has been described, symptoms mostly manifest after late childhood or in adults [88]. 

Also belonging to the group of PNPLA6-related disorders is Gordon Holmes syndrome. Many of the symptoms of Gordon Holmes patients overlap with the ones described in Boucher–Neuhäuser patients, however they do not show chorioretinal dystrophy [99,100,101,102]. A prominent feature in Gordon Holmes syndrome is hypogonadotropic hypogonadism and experiments treating a mouse pituitary cell line with an OP suggest that the effects on sexual development are due to a decreased release of luteinizing hormone from the pituitary glands [99]. Again, the mutations associated with this syndrome are recessive, and so far, have been localized to the C-terminal half of the protein.

The list of PNPLA6-related disorders was further expanded in 2015, with the discovery that mutations in PNPLA6/NTE cause both Oliver–McFarlane syndrome and Laurence–Moon syndrome. Although symptoms in these patients can include hypogonadism and ataxia, the distinguishing feature is severe chorioretinal dystrophy/retinitis pigmentosa often occurring in early childhood [8,59]. Whereas the phenotypes in Oliver–McFarlane patients include congenital trichomegaly, this is absent in Laurence–Moon patients. Like the forementioned diseases, Oliver–McFarlane and Laurence–Moon syndrome are rare inherited recessive diseases. Although most of the mutations are localized to the esterase/phospholipase and cyclic nucleotide binding domains, others are outside these identified domains [8,59,103,104,105]. 

As described above, disease-causing mutations are not restricted to the phospholipase domain (or delete the phospholipase domain due to a frame-shift) and it is therefore still unclear whether all these diseases are due to changes in the phospholipase function and effects on lipid metabolism. However, even mutations outside the phospholipase domain may interfere with the phospholipase function of PNPLA6/NTE. This concept is supported by findings that neither point mutations in the phospholipase domain nor point mutations in the cyclic nucleotide binding sites can restore LPC levels in the *sws* mutant [72]. On the other hand, measuring the hydrolase activity in fibroblasts from patients and non-symptomatic carriers revealed that phenotypically unaffected carriers can have similar or even significantly lower esterase activity than patients [59], suggesting that other changes in PNPLA6/NTE may contribute to the disease.

## 8. Future Directions

As discussed above, mutations in PNPLA6/NTE can cause a spectrum of inherited diseases that, although showing overlapping phenotypes, also manifest distinct features. However, a clear genotype-phenotype correlation has not been established and future studies are needed to determine specific functional consequences of the various mutations. Notably, the PKA-regulatory function and how it may contribute to PNPLA6-related disorders has not been studied. In *Drosophila*, overexpression of PKA-C3 (which binds to SWS), can induce neurodegeneration [9] whereas its loss results in male sterility [106]. PKA-C3 is a unique catalytic subunit that is more closely related to its mammalian homolog PRKX than to the other catalytic subunits in *Drosophila*. Interestingly, mutations in PRKX have also been linked with changes in fertilization and sexual development [107]. It is therefore conceivable that some PNPLA6/NTE mutations could affect PRKX activity and contribute to specific aspects of PNPLA6-related disorders, including key aspects of sexual development. However, it is also possible that the variations in phenotypes are due to genetic background effects and other genetic factors that modify the effects of a mutation. Due to these syndromes being quite rare, it will however be difficult to identify such factors. Lastly, in light of the relationship to OPIDN, it cannot be excluded that environmental factors play a role in the development of a specific syndrome. 

## Figures and Tables

**Figure 1 metabolites-12-00284-f001:**
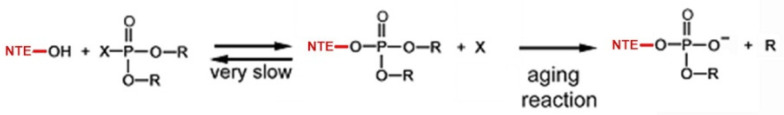
Interaction of neuropathic OPs with NTE. The hydroxyl group at the active site serine reacts with the phosphorus atom of the OP, resulting in the phosphorylation of NTE and the dephosphorylated OP (X). During the aging reaction, an R group from the phosphate moiety of NTE is released, resulting in a negative charge. Due to the aging reaction not being reversible, this results in the permanent inhibition of NTE’s esterase activity.

**Figure 2 metabolites-12-00284-f002:**
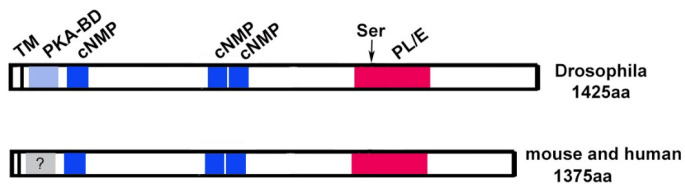
Schematic of PNPLA6/NTE and SWS. The catalytic domain, including the active site serine (arrowhead), is shown in red, the regions showing homology to PKA in blue. The transmembrane domain (TM) is indicated by a vertical line. PKA-BD=domain binding to the catalytic subunit (not confirmed in mammals), cNMP = cyclic nucleotide binding sites, PL/E = phospholipase/esterase domain.

**Figure 3 metabolites-12-00284-f003:**
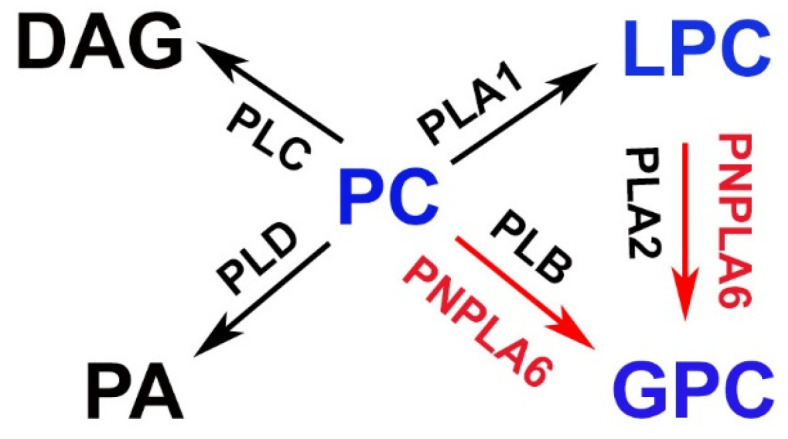
Proposed functions of PNPLA6/NTE in lipid metabolism. PNPLA6/NTE shows activity in hydrolyzing phosphatidylcholine (PC) or lysophosphatidylcholin (LPC) to glycerophosphocholine (GPC). DAG = diacylglycerol, PA = phosphatidic acid, PLA1 = Phospholipase A1, PLA2 = Phospholipase A2, PLB = Phospholipase B, PLC = Phospholipase C, PLD = Phospholipase D.

**Figure 4 metabolites-12-00284-f004:**
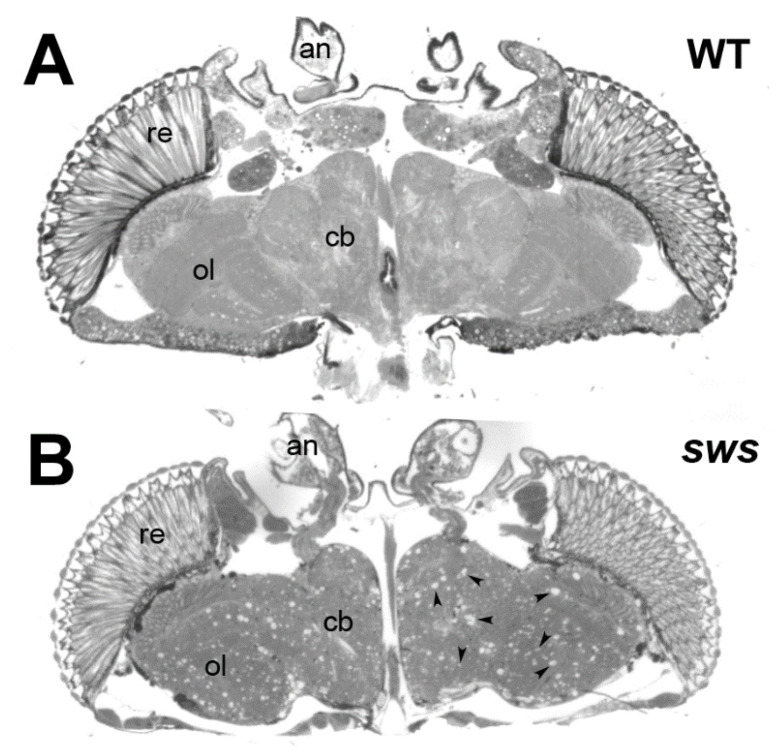
Loss of SWS results in neurodegeneration. (**A**) A horizontal head section from a 20d old wildtype *Drosophila* showing an intact brain. (**B**) In a head section from a 20d old *sws*^1^ mutant the neurodegeneration is visible by the formation of numerous sponfgiform lesions (some are indicated by the arrowheads) in all brain areas. re = retina, ol = optic lobes, cb = central brain, an = antennae.

**Figure 5 metabolites-12-00284-f005:**
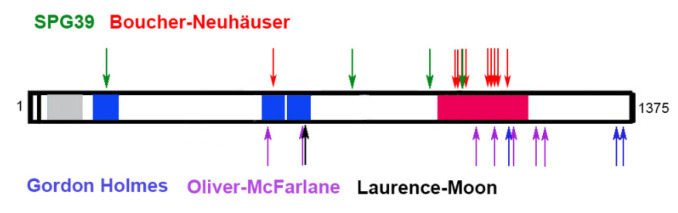
Disease-causing mutations in PNPLA6/NTE. Shown are only point mutations associated with the different syndromes. The cyclic nucleotide binding sites (aa195-322, aa511-633 and aa629-749) are shown in blue. The phospholipase domain (aa981-1147) is shown in red.

**Table 1 metabolites-12-00284-t001:** PNPLA6-related disorders. Characteristic clinical features are indicated by +. Clinical features not detected in all patients are indicated by +/−.

	Ataxia	Hypogonadism/ Delay in Sexual Development	Chorioretinal Dystrophy	Trichomegaly/ Alopecia	Spastic Paraplegia	Intellectual Disabilities	Dwarfism/ Short Statue
SPG 39	+/−				+		
Boucher–Neuhäuser	+	+	+/-		+/−	+/− mild	
Gordon Holmes	+	+			+/−	+/− mild	
Oliver–McFarlane	+/−	+	+ severe	+	+/−	+	+
Laurence–Moon	+/−	+	+		+/−	+	+

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
