# Peer review of "PNPLA6/NTE, an Evolutionary Conserved Phospholipase Linked to a Group of Complex Human Diseases"

_metabolites, 2022, doi:10.3390/metabo12040284_

Round 1

Reviewer 1 Report

This review presents the identification of 22 PNPLA6/NTE, its expression pattern, and normal role in lipid homeostasis, as well as the 23 consequences of altered NPLA6/NTE function in both model systems and patients. I recommend the acceptance in the present form.

Author Response

The reviewer had no comments

Reviewer 2 Report

The content of the manuscript is of scientific interest: the review discusses the structure, function, expression of PNPLA6/NTE and its orthologues. Indeed, despite the rather intensive studies carried out, in particular on model organisms, the functions of PNPLA6/NTE remain unclear. The review briefly analyzes almost all currently known works on PNPLA6/NTE and presents, in the author's opinion, further areas of research that can identify pathogenetic factors leading to the development of PNPLA6-related disorders. The material in the review is well structured and illustrated. Although the author has done a remarkable job,  taking into account the comments below, by my opinion, would improve the manuscript notably:

  1. Page 8, Line 300: (Dutta, Rieche et al. 2016) replace with [65].
  2. In the section "Phenotypic consequences due to the loss of PNPLA6/NTE in model systems", by my opinion, there is not enough information about studies conducted in rats and chickens. Although these were mainly toxicological studies, at least briefly they should be presented in this section.
  3. PNPLA6-related disorders are rare diseases, but in recent years the number of detected cases in various populations has been steadily increasing. It would be correct to add to the review links to new data that were published, perhaps when the work on the manuscript was already completed.

Chung EJ, You E, Oh SH, Seo GH, Chung WY, Kim YJ, Kim SJ.The First Korean Family With Boucher-Neuhauser Syndrome Carrying a Novel Mutation in PNPLA6. J Clin Neurol. 2022 Mar;18(2):233-234. doi: 10.3988/jcn.2022.18.2.233.

Nanetti L, Di Bella D, Magri S, Fichera M, Sarto E, Castaldo A, Mongelli A, Baratta S, Fenu S, Moscatelli M, Bonati MT, Martinuzzi A, Mariotti C, Taroni F. Multifaceted and Age-Dependent Phenotypes Associated With Biallelic PNPLA6 Gene Variants: Eight Novel Cases and Review of the Literature. Front Neurol. 2022 Jan 6;12:793547. doi: 10.3389/fneur.2021.793547. eCollection 2021.

He J, Liu X, Liu L, Zeng S, Shan S, Liao Z. Identification of Novel Compound Heterozygous Variants of the PNPLA6 Gene in Boucher-Neuhauser Syndrome. Front Genet. 2022 Feb 7;13:810537. doi: 10.3389/fgene.2022.810537. eCollection 2022

Author Response

  1. We have changed Dutta et al to [65].
  2. A discussion of the toxicology studies on hens and mice has been addet to section 6.
  3. We have added the novel references.

Reviewer 3 Report

In this review, the authors describe the identification of PNPLA6/NTE, its expression pattern, and normal role in lipid homeostasis, as well as the consequences of altered NPLA6/NTE function in both model systems and patients. Overall, the logic of the review is clear, and it comprehensively summarizes the progress of related research. Language expressions and charts of review are relatively standardized.

There are several suggestions for the authors to strengthen the quality of this review before the acceptance for publication.
(1) The roles of NPLA6/NTE possible included some other function, such as the differentiation of embryonic stem cells and blood vessel development.

(2)In the figure 4B, author should indicated the parts of “re,ol,cb,an” and neurodegeneration.

Author Response

(1) We have added a discussion of the functions of PNPLA in stem cells and embryogenesis to section 6.3.

(2) re, ol, cb, and an labels have been added to fig. 4B and arrowheads have been added to point to the neurodegenerative lesions